# Does Urban Agglomeration Discourage Entrepreneurship in China? Micro-Empirical Evidence from China

Wan Li [1], Bindong Sun [2,*], Shuaishuai Han [3] and Xiaoxi Jin [4]

1 Business School, Zhengzhou University, Zhengzhou 450001, China
2 Research Center for China Administrative Division, Institute of Eco-Chongming, Future City Lab, The Center for Modern Chinese City Studies, School of Urban and Regional Science, East China Normal University, Shanghai 200241, China
3 Key Research Institute of Yellow River Civilization and Sustainable Development & Collaborative Innovation Center of Yellow River Civilization Provincial Co-construction, Henan University, Kaifeng 475001, China
4 School of Urban and Regional Science, East China Normal University, Shanghai 200241, China
* Correspondence: bdsun@re.ecnu.edu.cn

**Abstract:** As the net effect of agglomeration on entrepreneurship depends on the trade-off between positive and negative effects, urban agglomeration can either promote or discourage entrepreneurial activity in theory. However, there is an unexpected shortage of empirical confirmations on this potential cause-and-effect relationship. Our study strives to fill this empirical gap by providing credible evidence whether agglomeration, measured by the urban density or population, increases the probability of individuals being self-employed. Based on the China Labor-Force Dynamic Survey of 2012, 2014, and 2016, we find that big cities fail to facilitate individuals to start or run their own businesses. Further analyses illustrate that the entrepreneurs in large cities can be easily tempted by a wider range of salaried opportunities and are generally exposed to high fixed costs and intense competition. In contrast, entrepreneurship in large cities is of high reward. These results serve as direct evidence of the co-existence of agglomeration diseconomies and economies. This also suggests the direction of government policy in large cities, which is to alleviate, as much as possible, the negative impact on entrepreneurs.

**Keywords:** agglomeration economies; agglomeration diseconomies; entrepreneurship; self-employment; agglomeration cost





## 1. Introduction

Entrepreneurship, the essence of which is creative destruction [1], does not only create employment and promote productivity, but also fundamentally affects cities' future evolution. A prominent phenomenon in China's entrepreneurial boom is the uneven geographical distribution of entrepreneurial activity. Beijing, Shanghai, Shenzhen, or other densely populated cities are often considered "pioneer cities of innovation and entrepreneurship in China" or "best cities of entrepreneurship in China". This is widely supported by the "China City Entrepreneurship Index" released by Renmin University of China (https://news.ruc.edu.cn/archives/126019, accessed on 14 December 2022) and the "Best Startup Cities in China" list issued by China's leading startup community (CYZONE) (https://www.cyzone.cn/article/132069.html, accessed on 14 December 2022). Given the highly spatial concentration of entrepreneurial activities, agglomeration economies are commonly considered as a starting point to understanding the generation and development of entrepreneurship [2–4]. Traditionally, cities with a large population or a high density have been regarded as "incubators" or "nurseries" for entrepreneurs [5,6]. Glaeser et al. (2010) also affirm that entrepreneurs in densely populated urban regions have the advantages of ready access to agglomerated local inputs, skills, ideas, and markets, among others.

However, there is a surprising lack of rigorous empirical evidence to test this assumed cause–effect relationship between urban agglomeration and entrepreneurship. There are a limited number of studies [7,8] that both used urban population size or density as their main variable of concern in the estimation of the effect of agglomeration on entrepreneurship and addressed endogeneity concerns. We consider that this absence of empirical verifications is due to the following issues. First, most researchers rarely question the positive effect of urban agglomeration. In previous empirical studies on the sources of entrepreneurship, while agglomeration has been covered, it has often been treated as a control variable [9–11]. Second, although these studies confirm that big/dense cities are friendlier to entrepreneurs, the causal relationship between these two variables remains questionable. The endogeneity problem has no easy treatment [12], as the main sources of endogeneity are sorting and potential omitting variables. We discuss these issues in more detail in the literature review section.

This paper, therefore, aims to provide a quantitative assessment on whether agglomeration, measured by urban density or population, increases the probability of individuals becoming entrepreneurs. With regard to entrepreneurship, there is no agreed measurement. We take respondents who claim they are self-employed as entrepreneurs, which is believed to be the most commonly used measurement of entrepreneurship [10]. Our study contributes to the literature in two important ways. First, it is one of the first quantitative attempts to establish the causal relationship between urban agglomeration and entrepreneurship. We use agglomeration, measured by urban density or population, as our focal variable and tackle the potential endogeneity problem by using a restricted subsample and two-stage least squares (TSLS) regressions. Our findings support the existence of agglomeration diseconomies and even suggest that the cost of agglomeration has surpassed its benefit in terms of entrepreneurship in China. Second, this paper identifies the source of our counter-intuitive finding, that is, why large cities fail to boost the possibility of being self-employed. Although previous studies have begun questioning the long-held positive effect of urban population size or density on entrepreneurship, there is still a shortage of evidence-based explorations in this area [7,13]. For an emerging market economy like China, which is in the process of institutional transformation and rapid urbanization, how to build and optimize urban entrepreneurial ecosystems is undoubtedly an issue worthy of attention in current research. The purpose of this paper is to explore the above-mentioned uneven geographical distribution of entrepreneurial activities from a relatively new perspective of urban agglomeration [14–16].

The paper is organized as follows. Section 2 lays out the theoretical background and progress on relevant empirical evidence. Section 3 describes the data and empirical strategy. Section 4 presents the econometric results and the final section concludes the paper.

## 2. Literature Review and Research Proposition

This section reviews the theoretical and empirical research on the relationship between urban agglomeration and entrepreneurship. Since it is a core topic in economic geography, there is a rich body of literature dedicated to agglomeration economies [3,5,12,14,17]. While it has long been established that the spatial concentration of firms and workers increases productivity, theoretically, the benefits of agglomeration accumulate faster initially, but eventually, costs prevail as population and density increase in cities [18,19]. Therefore, we next theoretically approach the effects of agglomeration on entrepreneurship from the benefit-cost perspective.

First proposed by Duranton and Puga (2004), agglomeration economies, or the benefits of agglomeration, are wildly widely recognized to stem from three sources: sharing, learning, and matching. Sharing means that the increased local outcomes of spatial concentration lie primarily in sharing indivisible facilities, input suppliers, industrial specialization, and risks, while learning suggests that the improvements in the local productivity of spatial agglomeration come largely from the generation, diffusion, and accumulation of knowledge. These two sources of agglomeration also motivate entrepreneurship, as the sharing

and learning effects in large cities are accelerators for entrepreneurs [20–22]. However, the matching mechanism of agglomeration economies may not serve the same function when it comes to entrepreneurship. Specifically, the boost in local performance from urban agglomeration mainly lies in the improvement of either the quality or quantity of the matches between firms and workers. On one hand, this helps entrepreneurs find employees and partners easily and efficiently, thus encouraging entrepreneurial activity; on the other hand, a higher matching effect in large cities also implies it is easier to find a satisfactory job, meaning individuals tend to become salaried-job employees rather than risk-taking employers.

In addition to this matching effect, there are other often mentioned costs of agglomeration, such as high land/house prices or intense competition, which are expected to negatively affect entrepreneurship [23]. The high land/house price costs are commonly believed to have a direct negative impact on entrepreneurship. Induced by agglomeration, high land/house prices suggest office or store rent required is likely higher for entrepreneurs in larger cities. Moreover, high land/house prices also mean entrepreneurs need to offer high salaries to enable their employees to afford rent. As for the intense competition, while some scholars argue that it makes entrepreneurship more efficient [7,14], others believe that excessive competition can discourage entrepreneurs [7]. Other costs, such as congestion, pollution, and crime, do not directly affect the profits or costs of entrepreneurial activity, and are thus not further discussed in this paper.

Apart from the theoretical uncertainty, empirical studies on the impact of urban agglomeration on entrepreneurship are lacking. There are limited empirical papers devoted to this specific topic [7,8] and their findings are inconsistent. Specifically, considering Italian college graduates' work possibilities as entrepreneurs after graduation, Di Addario and Vuri (2010) found that young college graduates were discouraged from starting their entrepreneurial activity in the most densely populated provinces. However, Sato et al. (2012) found that a U-shaped relationship existed between population density and observed entrepreneurship in Japanese prefectures, and the impact of population density on observed entrepreneurship was positive in both small and large cities, while the impact was smaller (or even negative) in medium-sized cities. While there are empirical studies on entrepreneurship that include urban agglomeration as control variables, these studies do not generally discuss the endogeneity of agglomeration and arrive at varied findings [10,11,24]. Similarly, there are empirical-based studies that focus on the industrial structure within agglomerations to explore the impact of specialization and diversification on entrepreneurship [25,26]. We do not further discuss these two branches of literature here because their topic is beyond the scope of this paper.

There are, in fact, two critical challenges in empirically answering the question of whether urban agglomeration increases the probability of an individual becoming an entrepreneur. They are also the main endogeneity sources. The first challenge refers to addressing the sorting or self-selection effect [7,14,27,28]. Specifically, both risk-taking entrepreneurs and risk-averse employees prefer to relocate to large cities because of the greater availability of both entrepreneurial and employment opportunities there. This re-location influences both the population size and level of entrepreneurship of a city, thus leading to biased estimates of the impact of agglomeration on entrepreneurship. Moreover, it is difficult to determine whether this is an overestimate or underestimate. However, this self-selection or sorting effect may not introduce a heavy bias. According to Michelacci and Silva (2007) [29], entrepreneurship can be regarded as a local factor, given that entrepreneurs tend to start their business in the regions they were born.

The second challenge regards missing variables [7,30,31]. To some degree, it is impossible for any study to rule out the possibility of missing variables. Attributes such as the cultural atmosphere of entrepreneurship are likely to influence both the urban population and its entrepreneurship level but are difficult to fully capture. This can lead to biased and inconsistent estimates of urban agglomeration, and ultimately to the failure to establish the causal link between agglomeration and entrepreneurship. It is also worth noting that,

while both studies deal with endogeneity using instrumental variables, neither pays special attention to the issue of self-selection.

Taken together, we make our research proposition as follows. It is difficult to conclude whether urban agglomeration promotes or discourages entrepreneurship, as the net effect of agglomeration on entrepreneurship depends on the trade-off between positive and negative influences. Notably, there is a good chance that agglomeration poses a disadvantage for entrepreneurship, with the potential disadvantages or agglomeration diseconomies being mainly embodied in alternative salaried opportunities, high land/house prices, and intense competition. Therefore, there is an urgent need for more empirical evidence to test this potential cause-and-effect relationship, while paying attention to endogeneity issues.

## 3. Data and Estimation Strategies

### 3.1. Data

Our main data source is the China Labor-Force Dynamic Survey (CLDS), which is a nation-wide database updated by Sun Yat-sen University every two years. The CLDS provides a representative image of China's workforce population and we focus on its 2012, 2014, and 2016 waves. Our sample consists of 11,551 working individuals (self-employer and employees), with a self-employment rate of about 17.98% (Table 1). The self-employment rate indicates that our data source is reliable, as it is consistent with the results of a sample survey of 1% of China's population. According to Wu et al. (2014) [32], which is based on the 2005 China's population sample survey (1/5 of a random subsample), the self-employment rate of the urban population was 13.1% in 2005. Generally, the self-employment rate is expected to remain stable and the rapid growth in 2014 mirrors the initiation of a policy on "mass entrepreneurship and innovation".

**Table 1.** Distribution of the sample between the self-employed and employees.

| Year | Self-Employer | | Employee | |
|------|-----|-----|-----|-----|
| | Number | Percentage (%) | Number | Percentage (%) |
| 2012 | 474 | 16.68 | 2368 | 83.32 |
| 2014 | 811 | 17.08 | 3938 | 82.92 |
| 2016 | 799 | 20.18 | 3161 | 79.82 |

Although there is no agreed measurement of entrepreneurship, self-employment is considered the most natural individual measure of entrepreneurship [10,33,34]. Hence, we start constructing the core explained variable Entrep, which is a dummy variable taking the value of 1 if the respondents state they are self-employed. Moreover, as a robust check, we also employ Active_Entrep, which is also a dummy variable taking the value of 1 if the respondents state that he or she was motivated to start a business based on taking advantage of a good business opportunity. The individuals who are self-employed as nannies or in odd jobs are dropped from the sample, as they are not really engaged in entrepreneurial activities. For further analysis, we also collect other information at the individual level (see Table 2).

Our core explanatory variable—urban agglomeration—is a proxy of urban density or population. The three CLDS waves considered in this study cover a total of 78 cities, providing a good national representation. A piece of supporting evidence is that the density and population distribution in our sample of 78 cities is similar to that of the national cities (Figure 1a,b). To be specific, the cities in our sample not only share a similar trend of density with all cities, but also have a wide population range, from 0.32 million (Yunfu) to 22.30 million (Shanghai). Moreover, these population-related data are all gathered from the 2010 Population Census of the People's Republic of China to ensure that the permanent population is considered. We also collect other city-level data from the China City Statistical Yearbook for the following analysis (Table 2).

**Table 2.** Variable definitions and summary statistics.

| Variable | Definition | Obs. | Mean | Std. Dev. |
|---|---|---|---|---|
| Entrep | Self-employed or not (1 = yes; 0 = no) | 9883 | 0.170 | 0.375 |
| Age | Age of the respondent (years) | 9883 | 40.475 | 10.492 |
| Male | Gender (1 = male; 0 = female) | 9883 | 0.554 | 0.497 |
| Edu_year | Years of schooling (years) | 9883 | 12.129 | 3.465 |
| Married | Marital status (1 = married; 0 = single) | 9883 | 0.821 | 0.384 |
| Income (ln) | Total income over the past year (Yuan, ln) | 9883 | 10.395 | 0.731 |
| Local_hukou | Possess a local hukou or not (1 = yes; 0 = no) | 9883 | 0.774 | 0.418 |
| Party | Being party member or not (1 = yes; 0 = no) | 9883 | 0.170 | 0.375 |
| Density (ln) | Population density (ln) | 78 | 9.602 | 0.487 |
| Population (ln) | Number of permanent population (ln) | 78 | 14.375 | 1.046 |
| Land area (ln) | Area of construction land (ln) | 78 | 4.772 | 1.063 |
| GDP_pop (ln) | GDP per capita (ln) | 78 | 10.329 | 0.534 |
| Coll_pop (ln) | Percentage of the total population with university education (ln) | 78 | −2.437 | 0.588 |
| Gov_gdp (ln) | Share of government expenditure (subtract expenditure on education and technology) in GDP (ln) | 78 | 7.249 | 0.418 |
| Ter_gdp (ln) | Share of tertiary sector output to GDP (ln) | 78 | 3.76 | 0.25 |
| Internet_pop (ln) | Number of international internet users per capita (ln) | 78 | −1.755 | 0.605 |

(ln) refers to the log-transformation of the data.

*3.2. Estimation*

To investigate whether urban agglomeration increases the probability of individuals being self-employed, we run the following logit regression:

$$Entrep = \alpha + \beta \ln(Density\ or\ Population) + \sum r_i ind_i + \sum \delta_j city_j + \varepsilon, \qquad (1)$$

where, as previously discussed, *Entrep* is a dummy variable indicating whether the respondent works as a self-employed entrepreneur, Density and Population are two continuous proxies for urban agglomeration, and $ind_i$ and $city_j$ are vectors of control variables at individual and city levels (*i* signifies different individuals, and *j* stands for different cities), respectively. Specifically, $ind_i$ includes respondents' age (Age), gender (Male), years of schooling (Edu_year), marital status (Married), income (Income), possessing a local hukou or not (Local_hukou), and being a party member or not (Party), while $city_j$ includes the area of constructed land (Land_area), GDP per capita (GDP_pop), city's average level of education (Coll_pop), share of government expenditure in GDP (Gov_gdp), share of tertiary sector output to GDP (Ter_gdp), and number of internet users per capita (Internet_pop). These control variables were primarily sourced from entrepreneurship and urban agglomeration studies [7,8,10,12,35,36]. We also include industry, province, and year fixed effects in the specification.

Although we focus on the three waves of the CLDS survey, we still employ a (pooled) cross-sectional strategy rather than panel regression. The main reason lies in the fact that a panel-based identification requires variation in the entrepreneurial status of individuals between 2012 and 2016 for weighing the impact of agglomeration on entrepreneurship. However, the entrepreneurs who have entered and exited the market are only around 200 during the study period (2012–2014). Therefore, our individual-level variables, including the core explained variables, cover three years (2012, 2014, and 2016), while all city-level variables and the core explaining variables are only for 2010. The definition and statistical information for all variables are outlined in Table 2.

To address the two empirical challenges mentioned above, we adopt two approaches. Our solution for the self-selection issue is to identify a subsample of only respondents that have not moved across counties since the age of 14. In this way, we rule out the risk-taking entrepreneurs and risk-averse employees who prefer to relocate among cities to some extent. Regarding potential missing variables, apart from adding region and year dummy

variables, we use a TSLS regression with an historical instrumental variable, that is, density or population in 1953, the data coming from the first census conducted in China.

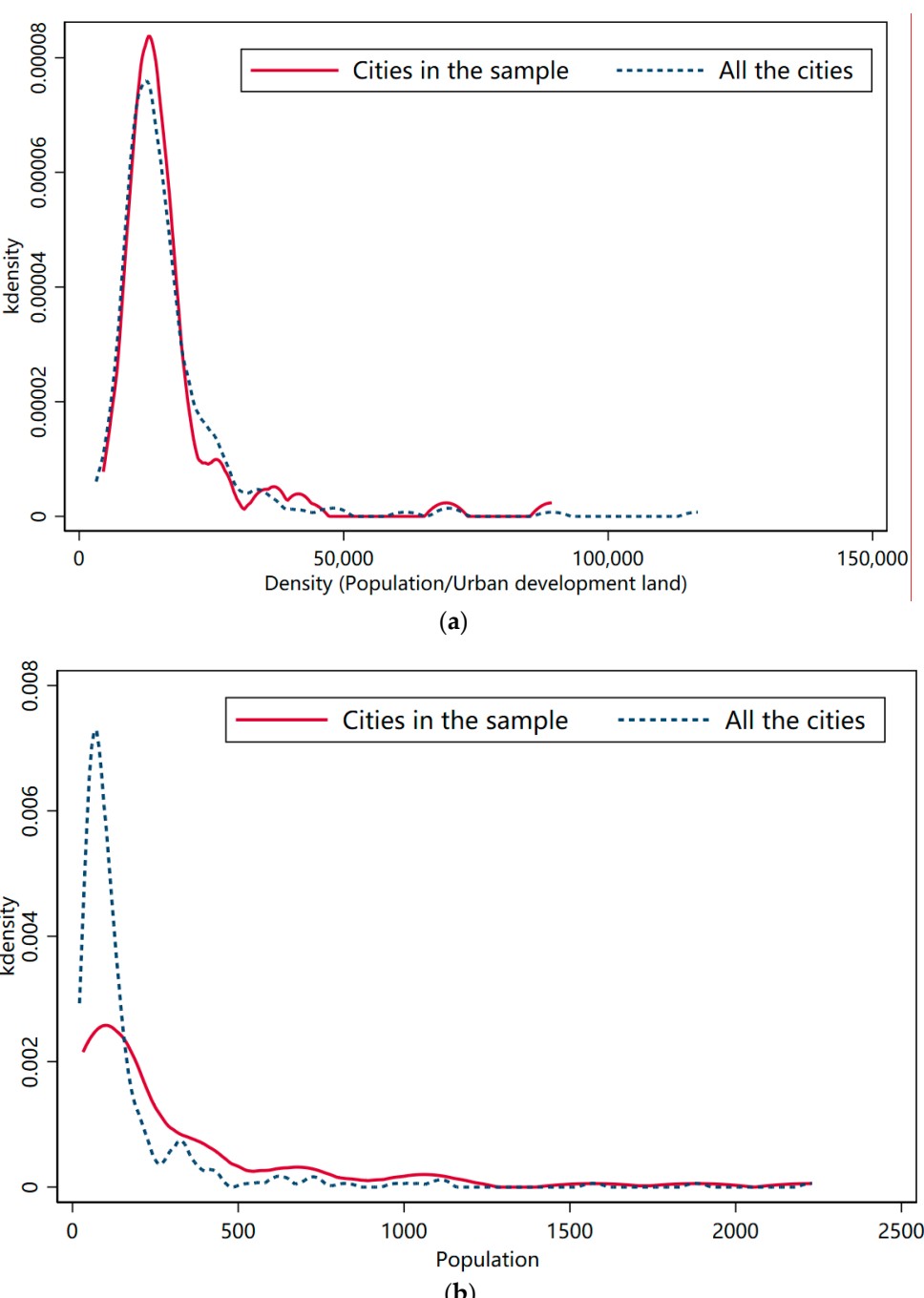

**Figure 1.** Kernel density estimate of urban density (**a**) and population (**b**).

Since the groundbreaking work of Ciccone and Hall (1996) [37], historical instrumental variables have become common practice in the study of agglomeration economies. This instrument can satisfy both the relevance and exogenous requirements. As for relevance, the population in 1953 shaped today's population. As shown in Table 3, the Kleibergen-Paap rk Wald F (KP F) statistic confirms the relevance of our instrument. As for exogeneity, the spatial pattern of China's population has changed dramatically in response to various broadly population-oriented projects, such as the well-known Shangshan Hsia-hsiang and The Third Front programs [38,39]. At the same time, entrepreneurship has also been deeply

transformed by three decades of planned economy. Hence, the 1953 population should have no direct effect on current entrepreneurship.

**Table 3.** Baseline identifications under different approaches.

| Y: *Entrep* | (1) MLE | (2) Sorting | (3) IV | (4) MLE | (5) Sorting | (6) IV |
|---|---|---|---|---|---|---|
| Density (ln) | 0.567 *** | 0.583 *** | 0.517 ** | | | |
| | (0.094) | (0.097) | (0.132) | | | |
| Population (ln) | | | | 0.567 *** | 0.583 *** | 0.517 ** |
| | | | | (0.094) | (0.097) | (0.132) |
| Age | 1.070 ** | 1.033 | 1.024 | 1.070 ** | 1.033 | 1.001 |
| | (0.030) | (0.043) | (0.023) | (0.030) | (0.043) | (0.002) |
| $Age^2$ | 0.999 * | 1.000 | 1.000 | 0.999 * | 1.000 | 1.000 |
| | (0.000) | (0.000) | (0.000) | (0.000) | (0.000) | (0.000) |
| Male | 1.465 *** | 1.810 *** | 1.388 *** | 1.465 *** | 1.810 *** | 1.008 |
| | (0.115) | (0.232) | (0.107) | (0.115) | (0.232) | (0.005) |
| Edu_year | 0.895 *** | 0.874 *** | 0.922 *** | 0.895 *** | 0.874 *** | 0.922 *** |
| | (0.013) | (0.022) | (0.015) | (0.013) | (0.022) | (0.015) |
| Married | 1.588 *** | 1.781 *** | 1.288 ** | 1.588 *** | 1.781 *** | 0.987 |
| | (0.208) | (0.368) | (0.154) | (0.208) | (0.368) | (0.154) |
| Local_hukou | 0.628 *** | 0.639 *** | 0.775 ** | 0.628 *** | 0.639 *** | 0.775 ** |
| | (0.089) | (0.105) | (0.080) | (0.089) | (0.105) | (0.015) |
| Income (ln) | 1.806 *** | 1.691 *** | 1.341 *** | 1.806 *** | 1.691 *** | 1.341 *** |
| | (0.188) | (0.194) | (0.091) | (0.188) | (0.194) | (0.091) |
| Party | 0.391 *** | 0.375 *** | 0.620 *** | 0.391 *** | 0.375 *** | 1.002 |
| | (0.063) | (0.062) | (0.062) | (0.063) | (0.062) | (0.006) |
| Land area (ln) | 0.696 *** | 0.731 ** | 0.673 ** | 1.228 | 1.254 * | 1.322 ** |
| | (0.091) | (0.094) | (0.114) | (0.153) | (0.155) | (0.212) |
| GDP_pop (ln) | 0.498 ** | 0.760 | 1.125 | 0.498 ** | 0.760 | 1.646 ** |
| | (0.158) | (0.238) | (0.296) | (0.158) | (0.238) | (0.296) |
| Coll_pop (ln) | 0.962 | 0.710 | 0.704 | 0.962 | 0.710 | 0.704 |
| | (0.240) | (0.194) | (0.189) | (0.240) | (0.194) | (0.189) |
| Gov_gdp (ln) | 0.588 ** | 1.098 | 0.867 | 0.588 ** | 1.098 | 0.867 |
| | (0.152) | (0.336) | (0.214) | (0.152) | (0.336) | (0.214) |
| Ter_gdp (ln) | 1.948 | 0.877 | 1.448 | 1.948 | 0.877 | 3.317 *** |
| | (0.954) | (0.424) | (0.823) | (0.954) | (0.424) | (0.823) |
| Internet_pop (ln) | 0.974 | 1.223 | 1.011 | 0.974 | 1.223 | 0.923 |
| | (0.237) | (0.299) | (0.206) | (0.237) | (0.299) | (0.206) |
| Industry FE | YES | YES | YES | YES | YES | YES |
| Province FE | YES | YES | YES | YES | YES | YES |
| Year FE | YES | YES | YES | YES | YES | YES |
| KP F stat | - | - | 9.961 | - | - | 9.961 |
| Observations | 9883 | 4266 | 3485 | 9883 | 4266 | 3485 |
| LL | −3163 | −1425 | 510.7 | −3163 | −1425 | 510.7 |
| Pseudo $R^2$ | 0.297 | 0.293 | / | 0.297 | 0.293 | / |

Odds ratio (OR) coefficients above 1 indicate an increased occurrence of the event and vice versa. Standard errors adjusted for clustering at city level are between parentheses. * $p < 0.1$, ** $p < 0.05$, *** $p < 0.01$.

## 4. Empirical Evidence

### 4.1. Baseline Results of Urban Agglomeration on Entrepreneurship

Table 3 reports the logit specification results of urban agglomeration on entrepreneurship for different proxy variables and econometric approaches. Columns (1)–(3) use the proxy variable Density, while columns (4)–(6) use Population as a proxy. Each column group uses the maximum likelihood estimate (MLE), sorting, and instrumental variable (IV) strategies, respectively. MLE is the most commonly used estimation strategy in logistic regression. Sorting refers to our adoption of subsamples that have not moved across counties since the age of 14 to tackle the potential sorting problem. IV implies estimation using TSLS with the 1953 density or population as IVs to address the missing variable concern.

Moreover, in the IV specification, the KP F statistic of columns (3) and (6) is close to 10, which is above the 15% maximal IV size (8.96) in the Stock–Yogo weak instrument test. This confirms the relevance of our instrumental variables.

Turning to our focal variable, the odds ratios for both Density and Population are below 1; that is, all else being equal, the higher the population density or the larger the population, the less chance individuals have of becoming self-employed. This finding is robust to three different strategies and the odds ratios are of roughly the same magnitude. In other words, big cities fail to incentivize individuals to start or run their own business, and our concerns about self-selection and omitted variables do not make a significant difference. The magnitude and significance of Density and Population are the same. As this is an inevitable result when considering Land area as a control variable, we only employ Density as the proxy for urban agglomeration in the following.

The results of the other controls are in line with expectations. At the individual level, male, married, and higher income individuals are more likely to be entrepreneurs, while individuals with high education, who hold a local residence, and are party members are less likely to engage in an entrepreneurial venture. This is consistent with Cejudo García et al. (2020) [40]. At the city level, the roles of these variables are rather mixed. In general, government intervention is harmful to individual entrepreneurship, while the average education level in the city does not affect whether an individual chooses to be self-employed.

Someone may argue that self-employment is not an appropriate measure for entrepreneurship, for many people are pushed into self-employment. In order to tackle this potential issue, we select the self-employed entrepreneur who claims that their motivation is to take advantage of a good business opportunity, as explained variables (Table 4). According to Table 4, all else being equal, the odds ratios for Density or Population are again below 1. In other words, for active self-employed entrepreneurs, big cities still play a negative role.

**Table 4.** Robust check with active entrepreneur.

| Y: *Active_Entrep* | (1)<br>MLE | (2)<br>Sorting | (3)<br>IV |
|---|---|---|---|
| Density (ln) | 0.423 *** | 0.332 *** | 0.517 ** |
| | (0.131) | (0.106) | (0.132) |
| Individual-level control variables | YES | YES | YES |
| City-level control variables | YES | YES | YES |
| Industry FE | YES | YES | YES |
| Province FE | YES | YES | YES |
| Year FE | YES | YES | YES |
| KP F stat | - | - | 9.961 |
| Observations | 5818 | 3640 | 3485 |
| LL | −1023 | −585.5 | 510.7 |
| Pseudo $R^2$ | 0.285 | 0.300 | / |

Odds ratio (OR) coefficients above 1 indicate an increased occurrence of the event and vice versa. Standard errors adjusted for clustering at city level are between parentheses. ** $p < 0.05$, *** $p < 0.01$. The individual-level control variables and city-level control variables in columns (1) and (2) are the same as in Table 3.

### 4.2. Potential Explanations for the Negative Impact of Agglomeration

As shown above, the probability of becoming an entrepreneur decreases as urban density increases. This result is robust for controlling for a wide range of individual- and city-level features and after correcting for the two potential endogenous sources of agglomeration. Here, we explore the potential explanations for our counter-intuitive finding from three aspects. Additionally, as the endogeneity problem is largely insensitive according to the benchmark regressions, in the follow-up specifications, we do not specifically target endogeneity to avoid MLE non-convergence. This is always the case, especially when the sample size is small. The sample size is reduced in many of the specifications in this section.

### 4.2.1. Matching Effect

It is believed that an increase in density or population can increase the probability of finding a match and improves the quality of matches. This translates into easier access to a satisfying job in dense or large cities; as a result, individuals tend to be wage-earning employees rather than risk-taking employers.

To verify this reasoning, we first divide our sample into high and low groups based on the availability of employment opportunities to check whether the magnitude and significance of Density differ. Next, we replace the explained variables with the job satisfaction to examine differences in match quality. Particularly, the availability of employment opportunities is measured by the total number of workers in the 2008 Industrial Census, while job satisfaction (ranging from 1 to 5 for strongly dissatisfied to strongly satisfied, respectively) is derived from the CLDS questionnaire.

We find that the coefficient on Density is significantly lower than 1 in column (1) but insignificant in column (2) of Table 5. This implies that the negative effect of agglomeration is stronger in cities with a higher availability of employment opportunities. Meanwhile, according to column (3), higher urban density is indeed associated with higher job satisfaction among employees. However, the increase in density has no significant effect on employer job satisfaction, based on column (4). This reflects the high quality of matches in large cities, where employees are more likely to find desirable jobs. In short, the negative effect of density can be explained by the fact that the densest markets are better at matching quantity with quality, thus creating a trade-off for entrepreneurship.

**Table 5.** Agglomeration economies and entrepreneurship: testing matching effect.

| | **(1)** | **(2)** | **(3)** | **(4)** |
|---|---|---|---|---|
| | **Y: *Entrep*** | | **Y: *Job Satisfaction*** | |
| | **High** | **Low** | **Employee** | **Employer** |
| Density (ln) | 0.207 *** | 1.199 | 1.188 ** | 1.003 |
| | (0.116) | (0.619) | (0.087) | (0.078) |
| Individual-level control variables | YES | YES | YES | YES |
| City-level control variables | YES | YES | YES | YES |
| Industry FE | YES | YES | YES | YES |
| Province FE | YES | YES | YES | YES |
| Year FE | YES | YES | YES | YES |
| Observations | 6753 | 2725 | 8135 | 1645 |
| LL | −2071 | −989.6 | −8393 | −1779 |
| Pseudo $R^2$ | 0.276 | 0.337 | 0.0360 | 0.0434 |

High signifies that the availability of jobs in this subsample is higher than the 50th percentile for the sample cities, while Low indicates it is below the 50th percentile. The individual-level control variables in columns (1)–(4) are the same as in Table 3. The city-level control variables in columns (1) and (2) are the same as in Table 3, but columns (3) and (4) include only GDP_pop and Land Area. OR coefficients are shown. Standard errors adjusted for clustering at the city level are between parentheses. ** $p < 0.05$, *** $p < 0.01$.

### 4.2.2. High House Price

As a non-tradable resource, land and housing prices are bound to increase with density, which can impose a high fixed cost on entrepreneurs and raise entrepreneurship entry barriers. Hence, it is generally agreed that high land/housing costs are a strong discouragement to entrepreneurship. Additionally, a side-effect of the high land/prices is that entrepreneurs typically have to pay high salaries to make house rent affordable for their employees. This may make the cost of labor additionally hinder entrepreneurship.

To explore whether this is the case, we split the sample into two high-low groups based on average house prices and salary in the city (Table 6). The coefficient on Density with high house prices is significantly below 1 but does not show significance for the subsample of low house prices. This empirically confirms the discouraging effect of house prices on entrepreneurship. In terms of salary, although the coefficient on Density with low labor cost is greater than 1, it is insignificant. In fact, none of the coefficients on density are significant

when grouped by salary (columns (3) and (4)). In other words, the dampening effect of labor costs is not verified. Overall, high housing prices in big cities tend to discourage individuals from entrepreneurship.

**Table 6.** Agglomeration and entrepreneurship: testing the effect of house price and salary.

| Y: *Entrep* | (1) House Prices | (2) | (3) Salaries | (4) |
|---|---|---|---|---|
| | High | Low | High | Low |
| Density (ln) | 0.644 * | 0.686 | 0.605 | 1.235 |
| | (0.166) | (0.373) | (0.232) | (0.364) |
| Other control variables | YES | YES | YES | YES |
| Industry FE | YES | YES | YES | YES |
| Province FE | YES | YES | YES | YES |
| Year FE | YES | YES | YES | YES |
| Observations | 6442 | 3082 | 6471 | 3066 |
| LL | −1977 | −1125 | −1986 | −1115 |
| Pseudo $R^2$ | 0.292 | 0.303 | 0.275 | 0.325 |

High signifies that the house price or salary in this subsample is higher than the 50th percentile for the sample cities, while Low indicates it is below the 50th percentile. The other control variables in columns (1)–(4) are the same as in Table 3. OR coefficients are shown. Standard errors adjusted for clustering at the city level are between parentheses. * $p < 0.1$.

### 4.2.3. Intense Competition

As for competition, it is widely accepted that there exists higher competition in larger markets. This is also confirmed by our empirical examination in column (1) of Table 7. Taking the employer's personal perception of intense business competition in the past year (Fierce, ranges 1 to 5 for free of competition to fierce competition, respectively) as the explanatory variable, an ordered logistic regression shows that the higher the density, the more intense perceived competition is.

**Table 7.** Agglomeration economies and entrepreneurship: testing the effect of competition.

| | (1) | (2) | (3) |
|---|---|---|---|
| | Y: *Fierce Competition* | Y: *Entrep* | |
| | | High | Low |
| Density (ln) | 2.006 *** | 0.734 * | 0.562 |
| | (0.511) | (0.118) | (0.346) |
| Other control variables | YES | YES | YES |
| Industry FE | YES | YES | YES |
| Province FE | YES | YES | YES |
| Year FE | YES | YES | YES |
| Observations | 999 | 3117 | 6476 |
| LL | −1159 | −1137 | −1967 |
| Pseudo $R^2$ | 0.059 | 0.296 | 0.299 |

High signifies that the density of firms in this subsample is higher than the 50th percentile for the sample cities, while Low indicates it is below the 50th percentile. The other control variables in columns (1)–(3) are the same as in Table 3. OR coefficients are shown. Standard errors adjusted for clustering at the city level are between parentheses. * $p < 0.1$, *** $p < 0.01$.

More importantly, fierce competition may invariably increase the difficulty of starting a business, which in turn discourages over-thinking entrepreneurs. Hence, we measure the degree of competition using the density of firms from the 2008 Industrial Census and divide the sample into high and low competition groups. As shown in Table 7, the subsample with a high competition degree has a regression coefficient on density significantly below 1 (column (2)), while the coefficient is insignificant for a low competition degree (column (3)). Therefore, the fierce competition in big cities does, as expected, drive individuals away from becoming employers.

### 4.3. Reward for Entrepreneurs in Large Cities

Based on Sections 4.1 and 4.2, we can conclude that large cities fail to encourage individuals to start or run their own businesses because entrepreneurs in large cities can easily be tempted by a wider range of salaried opportunities and face high fixed costs and intense competition. However, these findings can easily be translated into misleading policy, that is, limiting or restricting individuals from engaging in entrepreneurship in large cities. In fact, if a firm or entrepreneur can survive high housing prices and fierce competition in large cities, they can expect to reap significant rewards. In this sense, it is worth encouraging entrepreneurship in big cities.

To find out whether this is true, we respectively take the gross profit of firms, number of employees, and operational income of entrepreneurs as dependent variables and observe the coefficients on density. Table 8 confirms that, as urban density increases, firms and entrepreneurs indeed perform better. This can serve as a friendly reminder that the firms and entrepreneurs surviving in large cities are productive and do receive rewards.

**Table 8.** Rewards for entrepreneurs in large cities.

|  | (1) | (2) | (3) |
|---|---|---|---|
|  | Y: *Gross Profit of the Firm (Million RMB)* | Y: *Number of Employees* | Y: *Operational Income of Entrepreneurs* |
| Density (ln) | 0.006 * | 0.340 * | 0.500 *** |
|  | (0.003) | (0.184) | (0.125) |
| Other control variables | YES | YES | YES |
| Industry FE | YES | YES | YES |
| Province FE | YES | YES | YES |
| Year FE | YES | YES | YES |
| Observations | 1082 | 253 | 457 |
| $R^2$ | 0.161 | 0.356 | 0.299 |

Other control variables in columns (1)–(3) are the same as in Table 3. Standard errors adjusted for clustering at the city level are between parentheses. * $p < 0.1$, *** $p < 0.01$.

## 5. Discussion

Theoretically, it is difficult to draw conclusions on whether urban agglomeration promotes or hinders entrepreneurship. Based on our empirical examination, we find that, all else being equal, the higher the population density or the larger the population, the less chance individuals have of becoming self-employed. This baseline result is in line with Di Addario and Vuri (2010), who found that young Italian university graduates were reluctant to start their entrepreneurial activities in the most densely populated provinces. A U-shaped relationship was found in Japanese prefectures [8], but it could not be confirmed in our study (the square items of population and density are not statistically significant).

As for the reasons why large cities fail to encourage individuals to start their own businesses, we empirically find that entrepreneurs in large cities can easily be tempted by a wider range of salaried opportunities. This is against the suggestion of van Oort and Bosma (2013), who argue that the matching effect in large cities can help entrepreneurs find employees and partners easily and efficiently, thus encouraging the development of startups. Our findings lend more support to the idea that this matching effect makes it easier for individuals to find a satisfying job, which makes individuals tend to be employees with salaried jobs rather than risk-taking employers. Moreover, we find that entrepreneurs in large cities face intense competition. This is consistent with the Di Addario and Vuri (2010) argument about excessive competition.

Moreover, incredible rewards can be expected if a firm or entrepreneur can survive high housing prices and fierce competition in large cities. We do find firms in large cities are more likely to make better profits and hire more employees, and entrepreneurs can earn

higher incomes. This may explain why big cities have traditionally been seen as pioneering cities for entrepreneurship [41–43].

## 6. Conclusions

Cities with large or dense populations have traditionally been treated as entrepreneurial "incubators" or "nurseries" [44,45]. However, there is a surprising lack of rigorous empirical evidence to test this assumed cause–effect relationship between agglomeration economies and entrepreneurship. Based on the 2012, 2014, and 2016 CLDS waves, this paper tries to fill this empirical gap using credible specifications. We find that large cities fail to boost individuals to start or run their own businesses, and this primary finding is robust in correcting the two potential endogeneity sources of agglomeration. Further analyses illustrate that entrepreneurs in large cities can be easily tempted by a wider range of salaried opportunities and are largely exposed to high fixed costs and intense competition. Additional examinations find that firms in larger cities yield better profits and hire more employees, and entrepreneurs are more likely to have higher incomes.

These findings lead to critical implications for boosting entrepreneurship. First, our baseline finding is a timely reminder that the cost of agglomeration has even outweighed its benefit in terms of entrepreneurship in China. It can be further deduced that China's cities may be experiencing deviations from their optimal sizes. The most prominent agglomeration diseconomy is excessive housing prices, which pose a serious obstacle to entrepreneurship.

Second, our findings should not simply be reduced to the idea that we should limit or restrict individuals from engaging in entrepreneurship in large cities. Although a high density is a strong discouragement for individuals becoming entrepreneurs, the survivors in large cities can always expect significant rewards. According to our empirical examination, firms in large cities are more likely to make better profits and hire more employees, and entrepreneurs can earn higher incomes. The key message we aim to deliver is that entrepreneurs in large cities suffer from many disadvantages and mitigating these vulnerabilities is a top priority.

Third, targeted government policies to mitigate agglomeration diseconomies can focus on the following aspects. As entrepreneurs in large cities can be easily tempted by a wider range of salaried opportunities and are largely exposed to high fixed costs and intense competition, policymakers in large cities could at least nurture the culture of self-employment, reduce taxes for entrepreneurs, and encourage legitimate competition.

**Author Contributions:** Conceptualization, W.L., B.S., S.H. and X.J.; Data curation, X.J.; Formal analysis, W.L.; Funding acquisition, W.L., B.S. and S.H.; Supervision, B.S.; Validation, S.H.; Writing—original draft, W.L.; Writing—review & editing, B.S. All authors have read and agreed to the published version of the manuscript.

**Funding:** This research was funded by National Natural Science Foundation of China grant number 41901184, 42071210, Fundamental Research Funds for the Central Universities grant number 2022ECNU-XWK-XK001, Humanities and Social Science Youth Foundation of Ministry of Education of China grant number 22YJC790037, and Philosophy and Social Science Annual Project Youth Foundation of Henan Province of China grant number 2022CJJ130.

**Institutional Review Board Statement:** Not applicable.

**Informed Consent Statement:** Not applicable.

**Data Availability Statement:** Not applicable.

**Conflicts of Interest:** The authors declare no conflict of interest.

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
