# Peer review of "Does Urban Agglomeration Discourage Entrepreneurship in China? Micro-Empirical Evidence from China"

_land, doi:10.3390/land12010145_

Round 1

Reviewer 1 Report

This paper addresses urban agglomeration fuel entrepreneurial activity. The topic of the paper is interesting and authors in general show competent knowledge about the methodologies adopted. The conclusion of this paper is useful. However, there are some minor issues that need to be improved in the paper.

First, abstracts need to be better organized.

Second, the literature is not enough, the authors should add more related papers. The relationship between the reference and this paper needs to be further explained.

Third, the novelty of the paper should be better justified. I believe that the originality and contributions of the paper are related to the case-study but it should be underlined in a better way.

Fourth, Figure 1 is not complete, please replace the complete picture.

Other issues, please check the reference literature format to meet the journal requirements. Please check the grammar carefully, and use blank spaces when they are necessary.

 In sum, there is potential in this paper but it should be reviewed according to my recommendations and suggestions!

Author Response

We very much appreciate your positive feedback and constructive comments, which have helped us strengthen the overall quality of the manuscript. In response to your comments, we provide a point-by-point reply to illustrate how we have addressed your concerns.

Reviewer 2 Report

Dear Author(S), some details in attachment. 

Author Response

(The authors gave the same response as above.)
